# Gut Microbiota Signatures with Potential Clinical Usefulness in Colorectal and Non-Small Cell Lung Cancers

**DOI:** 10.3390/biomedicines12030703

**Published:** 2024-03-21

**Authors:** Sofía Tesolato, Juan Vicente-Valor, Mateo Paz-Cabezas, Dulcenombre Gómez-Garre, Silvia Sánchez-González, Adriana Ortega-Hernández, Sofía de la Serna, Inmaculada Domínguez-Serrano, Jana Dziakova, Daniel Rivera, Jose-Ramón Jarabo, Ana-María Gómez-Martínez, Florentino Hernando, Antonio Torres, Pilar Iniesta

**Affiliations:** 1Department of Biochemistry and Molecular Biology, Faculty of Pharmacy, Complutense University, 28040 Madrid, Spain; sofiteso@ucm.es (S.T.); juavicen@ucm.es (J.V.-V.); 2San Carlos Health Research Institute (IdISSC), 28040 Madrid, Spain; mateo.paz@salud.madrid.org (M.P.-C.); mgomezgarre@salud.madrid.org (D.G.-G.); ssgonzale@salud.madrid.org (S.S.-G.); a.ortega.hernandez@hotmail.com (A.O.-H.); sdlsernae@gmail.com (S.d.l.S.); inmaculadadominguezserrano@gmail.com (I.D.-S.); jana.dziakova@salud.madrid.org (J.D.); joseramon.jarabo@salud.madrid.org (J.-R.J.); agm912@hotmail.com (A.-M.G.-M.); fhtrancho@hotmail.com (F.H.); ajtorresgarcia@gmail.com (A.T.); 3Biomedical Research Networking Center in Cancer (CIBERONC), Carlos III Health Institute, 28029 Madrid, Spain; 4Cardiovascular Risk Group and Microbiota Laboratory, San Carlos Hospital, 28040 Madrid, Spain; 5Department of Physiology, Faculty of Medicine, Complutense University, 28040 Madrid, Spain; 6Biomedical Research Networking Center in Cardiovascular Diseases (CIBERCV), Carlos III Health Institute, 28029 Madrid, Spain; 7Digestive Surgery Service, San Carlos Hospital, 28040 Madrid, Spain; 8Department of Surgery, Faculty of Medicine, Complutense University, 28040 Madrid, Spain; 9Thoracic Surgery Service, San Carlos Hospital, 28040 Madrid, Spain

**Keywords:** microbiota, biomarker, colorectal cancer, non-small cell lung cancer

## Abstract

The application of bacterial metagenomic analysis as a biomarker for cancer detection is emerging. Our aim was to discover gut microbiota signatures with potential utility in the diagnosis of colorectal cancer (CRC) and non-small cell lung cancer (NSCLC). A prospective study was performed on a total of 77 fecal samples from CRC and NSCLC patients and controls. DNA from stool was analyzed for bacterial genomic sequencing using the Ion Torrent™ technology. Bioinformatic analysis was performed using the QIIME2 pipeline. We applied logistic regression to adjust for differences attributable to sex, age, and body mass index, and the diagnostic accuracy of our gut signatures was compared with other previously published results. The feces of patients affected by different tumor types, such as CRC and NSCLC, showed a differential intestinal microbiota profile. After adjusting for confounders, *Parvimonas* (OR = 53.3), *Gemella* (OR = 6.01), *Eisenbergiella* (OR = 5.35), *Peptostreptococcus* (OR = 9.42), *Lactobacillus* (OR = 6.72), *Salmonella* (OR = 5.44), and *Fusobacterium* (OR = 78.9) remained significantly associated with the risk of CRC. Two genera from the *Ruminococcaceae* family, *DTU089* (OR = 20.1) and an uncharacterized genus (OR = 160.1), were associated with the risk of NSCLC. Our two panels had better diagnostic capacity for CRC (AUC = 0.840) and NSLC (AUC = 0.747) compared to the application of two other published panels to our population. Thus, we propose a gut bacteria panel for each cancer type and show its potential application in cancer diagnosis.

## 1. Introduction

Non-small cell lung cancer (NSCLC) and colorectal cancer (CRC) are among the most common cancers in terms of incidence and are leading causes of death from malignant diseases worldwide. In 2020, there were 1.8 million and 0.9 million deaths due to NSCLC and CRC in the world, respectively [1,2]. Due to this high health burden, major efforts have been invested in research on prevention, diagnosis, prognosis, and treatment.

Several factors may play a role in the genesis and development of the tumorigenesis process. These are intrinsic factors related to individual characteristics, such as age and random mutations, and extrinsic factors, which depend on environmental features, such as pollutants, diet, and lifestyle [3]. The microbiota is one of the environmental factors that significantly influence the onset of cancer [4]. The vast quantity of microorganisms harbored by humans [5] can interact with host factors through metabolic, immunoinflammatory, and toxic pathways, driving tumor formation. For example, bile acid synthesis by *Clostridium* species has been shown to promote tumor growth [6], and short-chain fatty acids have been proven to protect against CRC [7]. In a mouse model of lung adenocarcinoma and in human samples, tumor tissue was enriched with γδ T cells in specific-pathogen-free mice (SPF) but not in germ-free (GF) mice [8]. *Fusobacterium nucleatum* expresses a bacterial cell surface toxin, FadA, an adhesion molecule that can activate β-catenin signaling by binding to E-cadherin, thereby promoting carcinogenesis [9].

Several experiments have shown that fecal microbiota transplantation from cancer patients to GF mice led to a greater number of tumoral lesions of larger size than in SPF mice bearing commensal microbiota [10]. However, other studies have concluded the opposite, where reduced commensal microbiota by antibiotics induced fewer tumor lesions [8]. These data support the need to conduct well-characterized studies in order to find generalizable biomarkers [11]. Additionally, some genera have been claimed to be increased in cancer in some studies but decreased in others, e.g., elevated [12] or reduced *Pseudomonas* [13] in tumor tissue from lung cancer patients. Therefore, human studies remain relevant to shed light on the development of microbial biomarkers for the diagnosis and prognosis of colorectal and lung cancer. The fecal microbiota has attracted the greatest interest in terms of biomarker identification, although the local microbiota from tumor and non-tumoral gut mucosa may also be of paramount relevance. Moreover, the gut microbiota can influence tumorigenesis at a distant site, such as the lungs. This finding has paved the way for the so-called gut–lung axis. According to this hypothesis, the gut microbiota can affect lung homeostasis, leading to increased susceptibility or resistance to lung diseases. Soluble microbial ingredients circulating via the gut–lung axis and interaction with dendritic cells and lung group 2 innate lymphoid cells would stand for the key interconnected means through which the gut microbiota participate in pulmonary diseases [14]. Taken together, these data seem to suggest that gut dysbiosis might contribute to CRC and NSCLC pathogenesis.

Recently, gut microbiome signatures have been developed across several cancer types [15,16]. However, whether there is a specific microbiome signature for each type of cancer, CRC or NSCLC, still remains under investigation. In CRC and NSCLC patients, a central core of pathogenic bacteria has been shown to be enriched, but the reality is that there has been a highly variable degree of species reported [15,17,18].

The main objective of this work was to compare gut microbiota profiles between different groups of subjects: CRC and NSCLC patients and control subjects. Furthermore, we intended to provide results to establish the relevance of the gut–lung axis. The analyses were performed on feces since this type of sample can be obtained non-invasively.

## 2. Materials and Methods

### 2.1. Patients and Samples

A total of 77 fecal samples from 38 CRC patients, 19 NSCLC patients, and 20 controls were collected and submitted to metagenomic analysis. Samples were obtained prospectively between 2021 and 2023. All cases came from the San Carlos Hospital in Madrid (Spain). Written approval to develop this study was obtained from the Clinical Research Ethics Committee of the San Carlos Hospital in Madrid (C.I. 19/549-E_BC, 27/12/2019). In addition, written informed consent was given by the subjects prior to investigation. All participants were recruited subsequently and regardless of gender, age, or tumor stage. Subject characteristics are depicted in Table 1. Patients were staged according to the American Joint Committee on Cancer classification [19] and all subjects were categorized according to their body mass index (BMI) values, following the criteria of the World Health Organization (WHO).

Eligible controls were voluntary subjects without cancer and considering the following exclusion criteria: previous history of inflammatory bowel disease; antibiotic treatment in the month prior to obtaining the stool sample; previous intestinal resection surgery; and oncological history, regardless of the time elapsed. In the cancer groups, eligible cases were patients undergoing potentially curative surgery for CRC or NSCLC at the San Carlos Hospital in Madrid. Cancer patients previously submitted to gastrointestinal resection surgery, those affected by inflammatory diseases, and those who had received antibiotic treatment one month before surgery were excluded. Moreover, chemo- and/or radiotherapy prior to the surgery was also considered an exclusion criterion.

Before the surgery or any systemic treatment, fresh fecal samples were collected and stored at −80 °C.

### 2.2. DNA Preparation and Sequencing

DNA was isolated from fecal samples using the QIAamp^®^ Fast DNA Stool Mini Kit (Qiagen, Hilden, Germany). DNA concentration was quantified using the Invitrogen™ Qubit™ 3 Fluorometer with the dsDNA HS (high sensitivity) Assay (Thermo Fisher Scientific, Waltham, MA, USA).

For microbiota analysis, amplification and sequencing of the 16S ribosomal RNA (rRNA) bacterial gene were conducted. The Ion Torrent™ sequencing technology and reagents from Life Technologies (a division of Thermo Fisher Scientific, Waltham, MA, USA) were employed, following previously established protocols [20]. Briefly, the DNA extracted from each sample (5 ng) underwent parallel polymerase chain reactions (PCRs). According to the Ion 16S™ Metagenomics Kit protocol, two distinct primer sets were used to amplify seven hypervariable regions of the gene (V2, V4, V8, and V3, V6–7, V9, respectively). Subsequently, the resulting amplicons were quantified, pooled equimolarly for each sample, and purified using Agencourt AMPure XP reagent beads (Beckman Coulter, Pasadena, CA, USA). Barcoded libraries were generated using the Ion Plus Fragment Library Kit and the Ion Xpress™ Barcode Adapters. After quantification using the Qubit™ 3 Fluorometer, libraries were set to 22 pM, pooled, and subjected to emulsion PCR with the Ion OneTouch™ 2 System and the Ion 520™ and 530™ Kit—OT2. Template-positive ion sphere particles were collected, washed, enriched using the Ion OneTouch™ ES Instrument OT2, and finally loaded onto an Ion 530™ Chip for sequencing with the Ion S5™ System.

### 2.3. Statistical Analysis

Bioinformatic analysis was conducted using the Quantitative Insight Into Microbial Ecology 2 (QIIME2) pipeline [21]. Sequences were assigned to operational taxonomic units (OTUs) at a 99% similarity threshold with SILVA 16S (v138). A minimum sampling depth of 100k reads per sample was applied for quality control.

Alpha diversity metrics, including observed OTUs, Chao1 richness estimate, Shannon diversity index, Pielou’s evenness index, and Simpson’s diversity index, were calculated from rarefied OTU profiles. Parametric tests (*t*-test) or non-parametric tests (Kruskal–Wallis test) were employed depending on data distribution for alpha diversity assessment. For beta diversity analysis, permutation-based multivariate analysis of variance (PERMANOVA), analysis of similarities (ANOSIM), and analysis of multivariate homogeneity of variances (PERMDISP) tests were conducted using the Jaccard and Bray–Curtis similarity indexes. Principal coordinate analysis (PCoA) was used for visualization.

Linear discriminant analysis effect size (LEfSe) [22] was used to identify taxa with significantly different relative abundances between groups. Taxa with a logarithmic LDA score (log10) > 2 and *p* value < 0.05 were considered statistically significant. The relative abundance values of taxa identified in the LEfSe analysis were further analyzed using the Cutoff Finder application [23]. This facilitated the establishment of relative abundance thresholds to distinguish between cancer patients and controls based on receiver operating characteristic (ROC) curve analysis, considering optimal area under the curve (AUC), sensitivity, and specificity values. Logistic regression was performed to adjust for potential confounding variables such as gender, age, and BMI differences.

Diagnostic parameters (AUC of ROC curve, sensitivity, and specificity) were calculated and compared between the identified fecal microbiota signatures and two large cancer microbiome signatures [15,16]. Statistical comparisons of ROC curves were conducted using the DeLong et al. method [24], with a significance threshold set at *p* value < 0.05. Statistical analyses were performed using STATA IC16.1 (Stata-Corp LLC, College Station, TX, USA) and IBM^®^ SPSS^®^ Statistics software package version 27 (IBM Inc., Armonk, NY, USA).

## 3. Results

### 3.1. Comparison of Microbial Diversity between Feces from CRC Patients, NSCLC Patients, and Control Group

Figure 1, Figure 2 and Figure 3 show the box plots and *p* values for the comparison of the alpha diversity metrics [Observed OTUs, (a); Chao1 index, (b); Shannon index, (c); Pielou’s evenness index, (d); and Simpson index, (e)], performed between feces from CRC patients and controls, NSCLC patients and controls, and CRC and NSCLC patients, respectively. When comparing the alpha diversity of microbiota between fecal samples from CRC patients and the ones from the control group, there were no significant differences. However, all metrics were higher in the CRC feces (Figure 1). A similar tendency was observed when the comparison was performed between feces from NSCLC patients and controls, with differences bordering statistical significance for the observed OTUs and the Chao1 index (Figure 2). Finally, the comparison of feces from the CRC patients and NSCLC patients reported no significant differences in alpha diversity, nor a particular tendency between both groups (Figure 3).

With respect to beta diversity, no significant differences between CRC or NSCLC patients and controls were detected, although in the NSCLC vs. controls comparison differences bordered statistical significance (in PERMDISP test, *p* value = 0.082 for the Jaccard index). However, when feces from the CRC and NSCLC patients were compared, differences in beta diversity became remarkable (in PERMANOVA test, *p* value = 0.010 for Bray–Curtis, and *p* value = 0.038 for the Jaccard index). Figure 4 shows bidimensional PCoA plots and *p* values for Jaccard beta diversity index between the study populations: CRC vs. control (Figure 4a), NSCLC vs. control (Figure 4b) and CRC vs. NSCLC (Figure 4c). 

### 3.2. Taxonomic Comparison of Microbiota between Feces from CRC Patients, NSCLC Patients, and Controls

Regarding phylum composition, all the groups shared *Firmicutes*, *Bacteroidota*, *Proteobacteria,* and *Actinobacteriota* as dominant taxa. However, the CRC and NSCLC groups displayed a decrease in phylum *Bacteroidota* when compared to the control group (CRC—NSCLC—control: 42.3% vs. 42.9% vs. 46.3%).

We next focused on the taxonomic differences between fecal microbiota from patients with CRC, patients with NSCLC, and control subjects at genus level. Figure 5 shows both the Venn diagrams and the LEfSe analyses at the bacterial genus level for the comparisons between fecal samples from CRC patients vs. controls (a), NSCLC patients vs. controls (b), and CRC patients vs. NSCLC patients (c). The *p* values corresponding to the bacterial genera featured in the LEfSe analysis are reported in Table 2.

As shown in Figure 5a, of the 1205 bacterial genera reported in the CRC vs. control comparison, 875 were shared by both groups, 271 were only present in the CRC group, and 59 were only present in the control group. LEfSe analysis at the bacterial genus level resulted in 17 genera differentially increased in CRC feces, 12 belonging to the *Firmicutes* phylum (*CAG-352*, *Lactobacillus*, *Ruminococcaceae*, *S5-A14a*, *Peptostreptococcus*, *Clostridium sensu stricto 1*, *Eisenbergiella*, *Gemella*, *Hydrogenoanaerobacterium*, *Turicibacter*, *Parvimonas,* and *Intestinimonas*), 3 to the *Proteobacteria* phylum (*Escherichia-Shigella*, *Eikenella,* and *Salmonella*), 1 to the *Fusobacteriota* phylum (*Fusobacterium*), and 1 to the *Campylobacterota* phylum (*Campylobacter*).

On the other hand, feces from the control subjects were enriched in five bacterial genera with respect to CRC patients, three from the *Firmicutes* phylum (*Blautia*, *Lachnoclostridium*, and *Dielma*), one from the *Proteobacteria* phylum (*Sutterella*), and one from the *Actinobacteriota* phylum (*Olsenella*).

When fecal samples from NSCLC patients were compared to the ones from the control group (Figure 5b), the Venn diagram reported 1085 genera, 812 in common between both populations, 138 only present in the NSCLC population, and 135 only present in the controls. LEfSe analysis indicated that 13 bacterial genera were significantly increased in feces from NSCLC patients. Five of these genera were common to the CRC group (*Turicibacter*, *Intestinimonas*, *Eisenbergiella,* and *Lactobacillus*, all belonging to the *Firmicutes* phylum; and *Salmonella*, from the *Proteobacteria* phylum), whereas the other eight genera were new for the NSCLC group (*Agathobacter*, *Ruminococcaceae Incertae Sedis*, *Granulicatella*, *Frisingicoccus*, *DTU089*, *Clostridium sensu stricto 3*, *Coprobacillus*, and *Hungatella*, all of them belonging to the *Firmicutes* phylum). Feces from controls were enriched in three genera with respect to NSCLC patients: genera *Lachnoclostridium* (phylum *Firmicutes*) and *Olsenella* (phylum *Actinobacteriota*), which were common to the CRC vs. control comparison, and genus *Megasphaera* (*Firmicutes* phylum).

Finally, the comparison of fecal samples from CRC and NSCLC patients (Figure 5c) reported 1017 bacterial genera, 804 of them shared between both cancer groups, 177 only present in CRC, and 36 only present in NSCLC. Regarding LEfSe analysis, feces from patients with CRC had 11 increased bacterial genera. Eight of these (*Fusobacterium*, from phylum *Fusobacteriota*; *CAG-352*, *Peptostreptococcus*, *Parvimonas*, *Hydrogenoanaerobacterium,* and *Ruminococcaceae*, all of them from phylum *Firmicutes*; *Eikenella*, from the phylum *Proteobacteria*; and *Campylobacter*, from phylum *Campylobacterota*) were already found to be increased in CRC patients with respect to the control group, whereas the other three genera (*Sarcina*, *Solobacterium,* and *Sellimonas*, all from phylum *Firmicutes*) were new for this comparison. Feces from NSCLC patients reported an increase in genera *Agathobacter* and *Ruminococcaceae Incertae Sedis* (phylum *Firmicutes*), which were also increased with respect to the control group, as well as in seven new bacterial genera (*Bifidobacterium*, phylum *Actinobacteriota*; *Roseburia*, *Blautia*, *Oscillibacter*, *Ammoniphilus,* and *Pseudobutyrivibrio*, all of them from phylum *Firmicutes*; and *CAG-873*, from phylum *Bacteroidota*).

Notably, when performing the taxonomic analysis between fecal samples from CRC patients with respect to tumor location and TNM stage, some of these genera also appeared to be increased in one of the groups. Particularly, genus *Campylobacter* was more abundant in feces from stage I or stage II CRCs (*p* value = 0.027 in LEfSe analysis), whereas *Ruminococcaceae* was associated with stage III-IV CRCs (*p* value = 0.033 in LEfSe analysis). Regarding primary tumor location, genus *Hydrogenoanaerobacterium* was increased in feces from rectal cancers with respect to CRCs from other locations (*p* value = 0.027 in LEfSe analysis), and both *Hydrogenoanaerobacterium* and *Fusobacterium* genera were more abundant in left colon cancers with respect to right colon ones (LEfSe *p* value = 0.021 for *Hydrogenoanaerobacterium* and *p* value = 0.048 for *Fusobacterium*).

### 3.3. Logistic Regression Analysis of Differentially Increased Bacterial Genera in Patients with CRC or NSCLC

A logistic regression analysis was performed on the bacterial genera that were found to be increased in either CRC or NSCLC patients with respect to the control group, in order to evaluate their potential as biomarkers for cancer development. First, the relative abundances of each taxon were categorized according to a threshold calculated by the Cutoff Finder application. Next, a logistic regression analysis was performed to identify the significant taxa after adjusting for sex, age, and BMI. The results from logistic regression in CRC and NSCLC patients can be seen in Table 3 and Table 4, respectively. As shown in Table 3, a higher abundance of genera *Parvimonas*, *Gemella*, *Eisenbergiella*, *Peptostreptococcus Lactobacillus*, *Salmonella,* and *Fusobacterium* was significantly associated with the presence of CRC when compared to controls. Genera *Campylobacter*, *Turicibacter*, *Clostridium sensu stricto 1*, *Eikenella,* and *Escherichia_Shigella* were also associated with CRC, although their logistic regression bordered the statistical significance. On the other hand, increased levels of genera *DTU089* or *Ruminococcaceae Incertae Sedis* were associated with the presence of NSCLC when compared to controls (Table 4), as well as genera *Hungatella*, *Clostridium sensu stricto 3*, *Salmonella,* and *Agathobacter*, which bordered statistical significance.

### 3.4. Comparison of Fecal Microbiota Panels

Based on the results of the LEfSe analysis, we performed a comparison of the diagnostic accuracy of a proposed bacterial panel for each cancer type (CRC or NSCLC), and the signatures from two large, published cancer databases. The details of this comparison are shown in Table 5. In both cases, the comparison of ROC curves yielded that our discovered signatures significantly outperformed the diagnostic ability of the previously published panels. The genera used for each panel can be found beneath Table 5. In the case of CRC, the proposed panel, and the one with best performance, included the seven bacterial genera increased in CRC feces with respect to control feces and of which the logistic regression was found to be significant (i.e., *Parvimonas*, *Gemella*, *Eisenbergiella*, *Peptostreptococcus, Lactobacillus*, *Salmonella*, and *Fusobacterium*). When applied to our population, this panel yielded an AUC of 0.840, a sensibility of 78.9%, and a specificity of 80%. The same criterion was applied for the NSCLC panel, comprised by two bacterial genera (*DTU089* and *Ruminococcaceae Incertae Sedis*). This panel performed with an AUC of 0.747, 73.7% of sensibility, and 75% of specificity, when applied to our population. ROC curves are displayed in Figure 6, where Figure 6a,b correspond to the performance in our study population of our proposed CRC and NSCLC panels, respectively, Figure 6c,d to the performance of the Thomas et al. published signature [15], and Figure 6e,f to the performance of the Yang et al. published signature [16].

Though the results shown are promising, an investigation in a larger cohort of patients is needed. Based on our data, we could now hypothesize a prevalence exposure in the control population of 45% for CRC (meaning that 45% of our control population had overrepresentation of at least one “hazardous” genus from our CRC signature). In the case of NSCLC, the prevalence exposure in our control population was 15% (meaning that 15% of our control population had overrepresentation of at least one “hazardous” genus from our NSCLC signature). Additionally, considering a minimum OR of association of 2, sample size for future studies considering CRC and NSCLC would be 266 (133 controls and 133 CRC patients) and 416 (208 controls and 208 NSCLC patients), respectively.

## 4. Discussion

In this work, we compared the fecal microbiota from patients affected by CRC or NSCLC with that from control subjects. These comparisons were based on the potential utility of fecal samples as a non-invasive way to detect pathologic changes in the gut microbiota and identify useful biomarkers for cancer diagnosis. The difference in age between the control group and the groups of patients affected by cancer could be, at least in part, a consequence of the exclusion criteria applied to the control subjects. It is not easy for an elderly patient not to meet any of the exclusion criteria that we established (young patients meet the necessary characteristics more easily). Additionally, in our case, CRC is mainly a pathology typically found in older patients, which probably explains the age differences.

Some previous studies have reported a decrease in alpha diversity in cancer patients, both at the tissue and the fecal level [25,26]. Controversially, our findings suggested a tendency to higher alpha diversity metrics in both cancer groups with respect to the controls. However, none of the differences observed was statistically significant, which is in line with other research [27]. Importantly, the observed microbiome differences usually involve relative quantitative differences in the abundance of specific taxa of bacteria [28].

A healthy gut microbiota is composed of a diverse group of commensal microorganisms, most of them belonging to the bacterial phyla *Firmicutes* and *Bacteroidota*, which help to maintain homeostasis by interacting with the host’s metabolism and immune system. In this context, both the pathogen infection and the infectivity of resident pathobionts are limited. Several diseases including different types of cancer have been linked to gut dysbiosis, which involves a shift to a more pathogenic microbiota. Feces from CRC and NSCLC patients included in the present study had a decrease in phylum *Bacteroidota*, when compared to feces from controls. Other studies [29] have also highlighted the decrease in this phylum in mucosal tissue as harmful.

At the genus level, our LEfSe analysis comparing feces from patients with CRC, patients with NSCLC, and controls revealed several bacterial genera which were enriched in each of the study groups. Putting the comparisons together, feces from both CRC and NSCLC patients shared five increased genera with respect to controls (*Intestinimonas*, *Turicibacter*, *Eisenbergiella*, *Lactobacillus,* and *Salmonella*), thus making them potential biomarkers for cancer. Moreover, CRC feces included eight genera (*Campylobacter*, *Parvimonas*, *Hydrogenoanaerobacterium*, *Peptostreptococcus*, *Ruminococcaceae*, *Eikenella*, *CAG-352,* and *Fusobacterium*) which seem to constitute a more specific CRC profile, as they were found to be differentially increased in this group of patients both when compared to controls and to NSCLC patients. On the other hand, genus *Agathobacter* and the unidentified genus *Ruminococcaceae Incertae Sedis* were increased in feces from NSCLC patients with respect to both control and CRC patients, thus being more specific for the profile of this type of cancer.

After adjusting the genera reported by LEfSe analysis for variables that could have influenced the composition of the microbiota, seven genera remained increased in CRC patients (*Parvimonas*, *Gemella*, *Eisenbergiella*, *Peptostreptococcus Lactobacillus*, *Salmonella,* and *Fusobacterium*), and two in NSCLC patients (*DTU089* and *Ruminococcaceae Incertae Sedis*), compared to controls. In CRC, an overrepresentation of *Fusobacterium*, *Porphyromonas*, *Parvimonas*, *Peptostreptococcus*, and *Gemella* can be found as an indicator of microbial dysbiosis [30]. Similarly, *Salmonella* [31] and *Eisenbergiella* [32] have also been linked to the risk of CRC. The role of *Lactobacillus* in the risk of developing CRC is more controversial, as the core scientific studies explore its benefits as a probiotic [33]. Some authors did not find differences in the abundance of *Lactobacillus* between CRC and controls [34]. The increase in *Lactobacillus* in our oncological population could be interpreted as reflecting a fight against the tumor process, a dysbiosis inherent to the tumor process itself, or a reflection of an incipient stage where the abundance of this particular bacterium has not been reduced. In NSCLC, an uncharacterized member from *Ruminococcaceae* family [35] was also found to be associated with NSCLC, as similarly were the uncharacterized genus *Incertae Sedis* and the genus *DTU089* in our study. Other bacteria genera that were close to statistical significance in our regression studies had also already been related to cancer in previous studies. Of the ones increased in CRC feces, *Campylobacter* spp., *Escherichia* spp., and *Shigella* spp., along with the already mentioned *Salmonella* spp., include pathogenic and toxin-producing bacteria that have been related to the progression of cancer [36]. *Turicibacter* spp. and *Clostridium sensu stricto 1* spp. were found to be positively correlated with the expression of several inflammatory cytokines as well as growth factor TGFb and transcription factor STAT3 in a CRC mouse model [37]. Finally, *Eikenella* spp. was suggested as passenger bacteria in CRC as it increased in tumors from CRC patients with respect to healthy control mucosa [38]. In the case of the bacteria found in NSCLC feces, overrepresentation of *Agathobacter* spp. and *Clostridium* spp. in feces from patients with this type of cancer had been previously reported [39]. *Hungatella* spp. was increased in feces from NSCLC patients with cachexia [40]. Finally, *Salmonella* spp. has been related to various types of cancer including CRC, gallbladder cancer, and hepatobiliary carcinoma, [41], although its role in lung cancer remains unclear. The mechanism by which the microbial taxa could be involved in cancer development is still the subject of thorough investigation. Whether the main actors are specific taxa or the dysbiosis as a whole is unknown; possibly, a combination of mechanisms could provide an explanation to the role of microbiota in carcinogenesis. Both types of imbalance would ultimately trigger epithelial–mesenchymal transition, inflammation with increased reactive oxygen species and DNA damage, and suppression of the immune response [42].

We then proposed two diagnostic bacterial panels for CRC and for NSCLC, based on the logistic regression results. When we compared the diagnostic performance of our augmented differential genera with that of previously published clusters, our panels displayed significantly better accuracy in predicting cancer status. The diagnostic accuracy of our proposed panels was considered acceptable (0.7 < AUC < 0.8) and very good (0.8 < AUC < 0.9) [43]. The relevance of the published bacteria panels for being as less applicable is uncertain for us, as their data come from a larger number of patients. It should be noted that the signature from Yang et al. [16] was developed for cancer prognosis, not with diagnostic purposes. However, both signatures previously published [15,16] showed no discriminatory performance. These results may reflect some lack of external validation. Therefore, the implementation of the microbiome features as a diagnostic cancer tool in clinical practice may need further research. It may be that a signature panel for diagnostic purposes cannot be applied with all the genera but only with a few of them. Moreover, considering that there are 3.8 × 10^13^ bacteria [44] in the human body and that we harbor over 10,000 species [45], this enormous biodiversity may defy the discovery of a universal oncomicrobiome signature. We may have to develop specific signatures for each continent, age range, cancer type, or even cancer stage.

As the main limitation of this work, we highlight the modest size of our population.

## 5. Conclusions

Our results indicate, as the main conclusion of the present study, that the feces of patients affected by different tumor types, such as CRC and NSCLC, show a differential intestinal microbiota profile. We propose a gut bacteria panel for each cancer type and demonstrate its potential application in cancer diagnosis. Furthermore, we report for the first time some bacteria associated with the cancer risk and perform a comparative analysis with bacteria panels coming from previous meta-analysis studies.

## Figures and Tables

**Figure 1 biomedicines-12-00703-f001:**
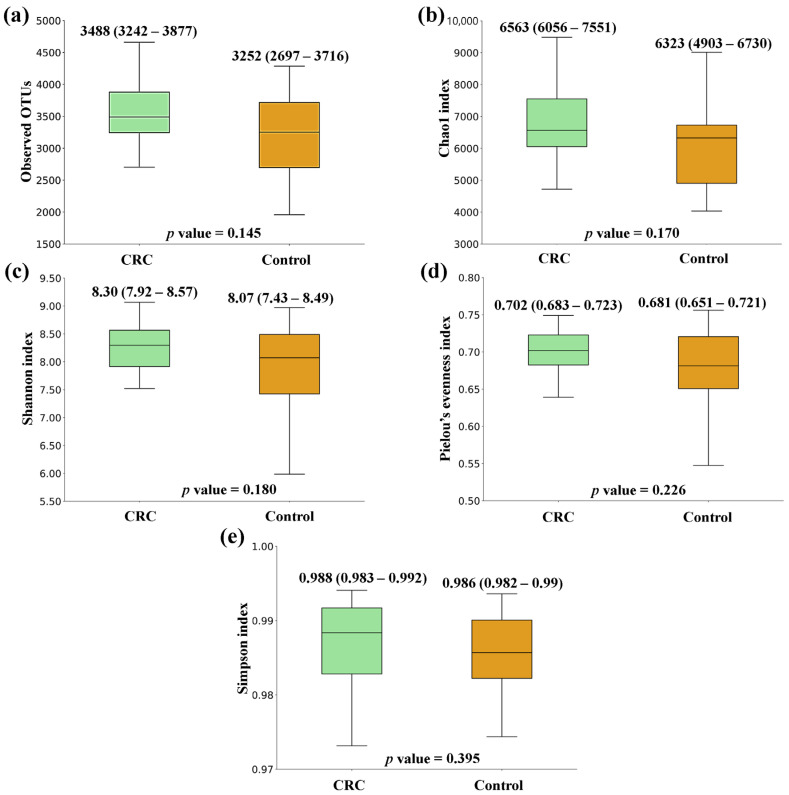
Alpha diversity comparison between feces from colorectal cancer (CRC) patients and controls. (**a**) Observed OTUs; (**b**) Chao1 index; (**c**) Shannon index; (**d**) Pielou’s evenness index; (**e**) Simpson index. Median values with interquartile range and *p* values are indicated.

**Figure 2 biomedicines-12-00703-f002:**
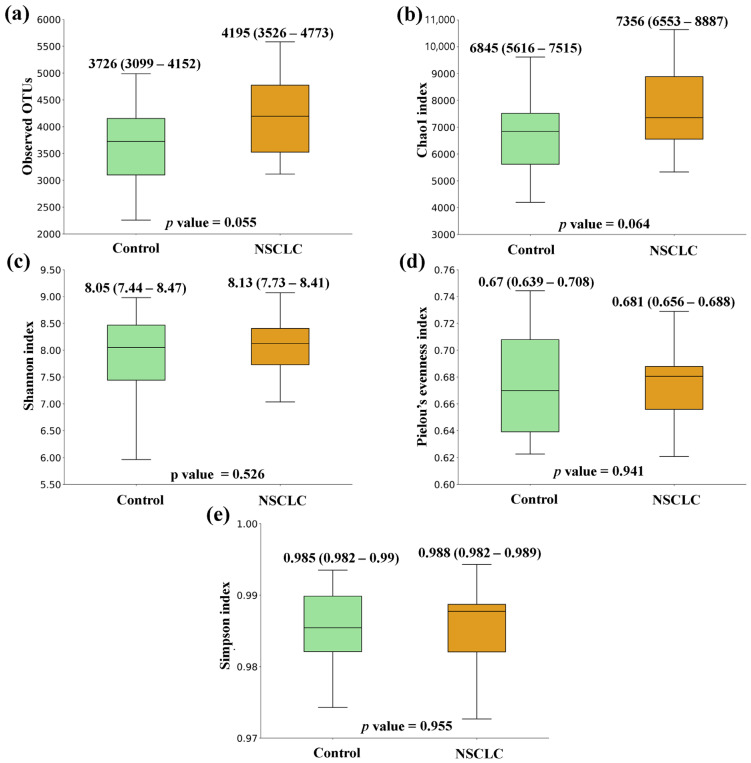
Alpha diversity comparison between feces from non-small cell lung cancer (NSCLC) patients and controls. (**a**) Observed OTUs; (**b**) Chao1 index; (**c**) Shannon index; (**d**) Pielou’s evenness index; (**e**) Simpson index. Median values with interquartile range and *p* values are indicated.

**Figure 3 biomedicines-12-00703-f003:**
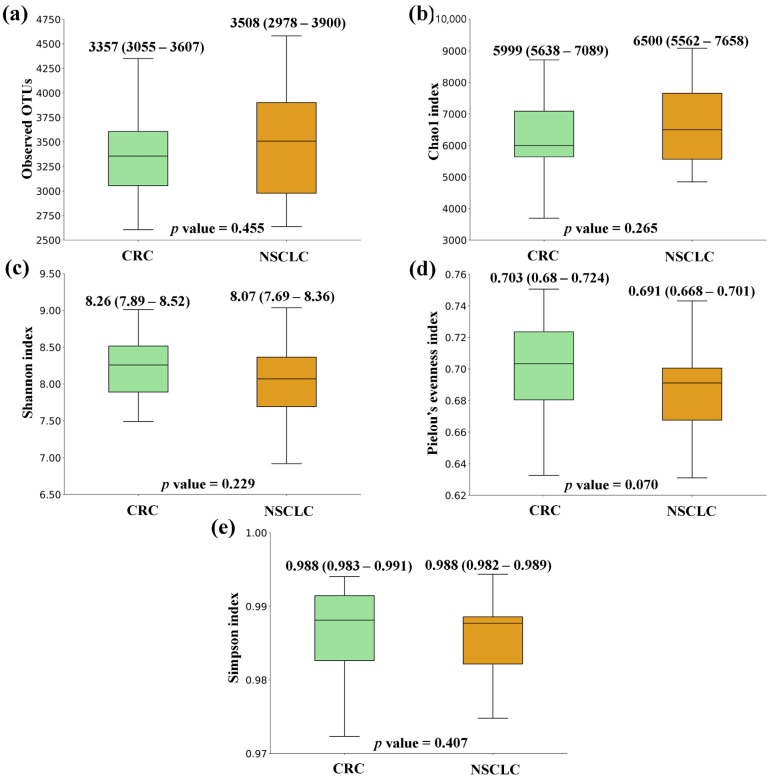
Alpha diversity comparison between feces from colorectal cancer (CRC) and non-small cell lung cancer (NSCLC) patients. (**a**) Observed OTUs; (**b**) Chao1 index; (**c**) Shannon index; (**d**) Pielou’s evenness index; (**e**) Simpson index. Median values with interquartile range and *p* values are indicated.

**Figure 4 biomedicines-12-00703-f004:**
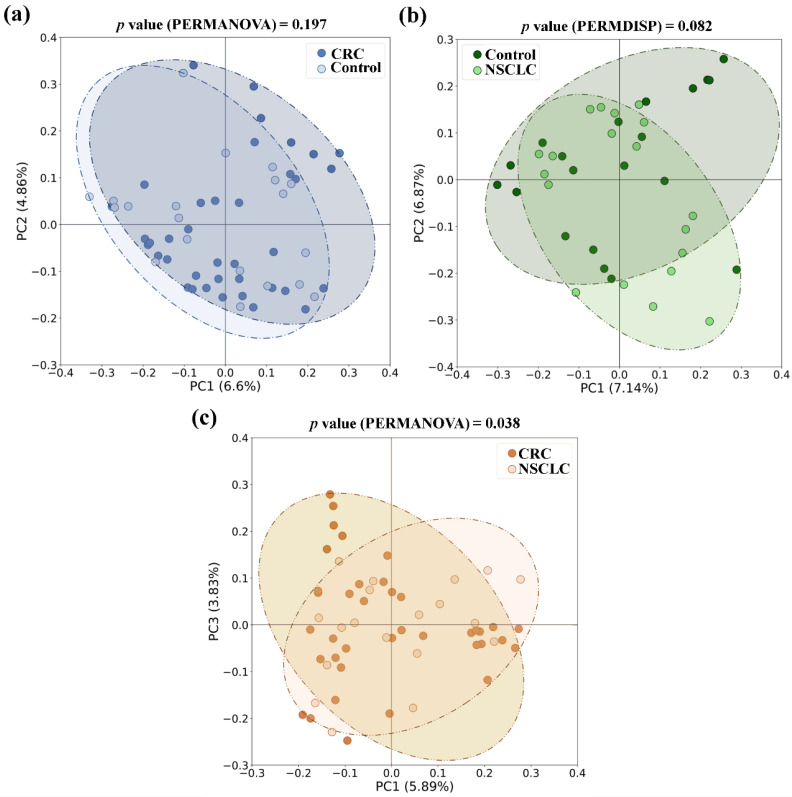
Principal coordinates analysis (PCoA) plots based on Jaccard index for feces from colorectal cancer (CRC) patients, non-small cell lung cancer (NSCLC) patients and controls. (**a**) CRC vs. controls (**b**) NSCLC vs. controls; (**c**) CRC vs. NSCLC.

**Figure 5 biomedicines-12-00703-f005:**
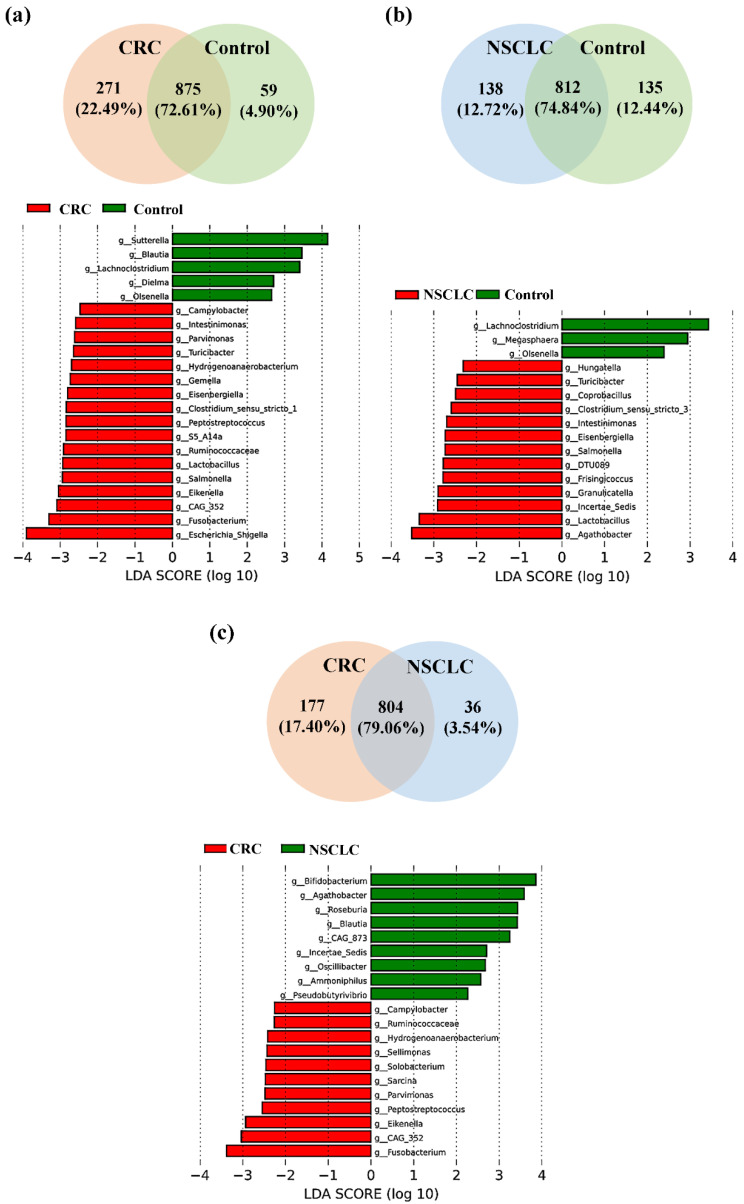
Taxonomic comparison at the bacterial genus level between feces from colorectal cancer (CRC) patients, non-small cell lung cancer (NSCLC) patients and controls (Venn diagrams and LEfSe analysis). (**a**) CRC vs. controls; (**b**) NSCLC vs. controls; (**c**) CRC vs. NSCLC.

**Figure 6 biomedicines-12-00703-f006:**
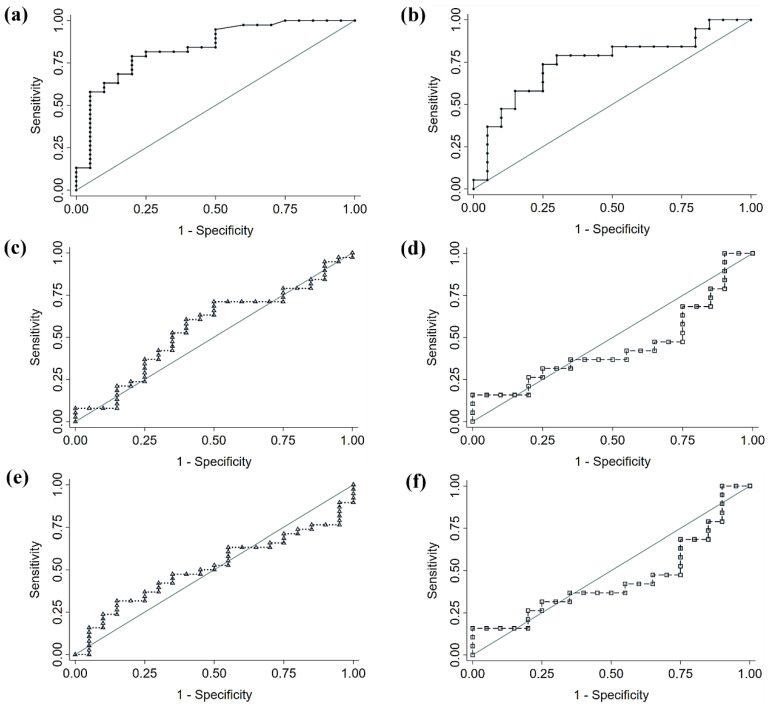
Receiver operating characteristic (ROC) curves of diagnostic accuracy of fecal microbiota signatures applied to our study population. (**a**) Proposed colorectal cancer (CRC) signature; (**b**) proposed non-small cell lung cancer (NSCLC) signature; (**c**,**d**) signature proposed by Thomas et al. [15] applied to CRC and NSCLC, respectively; (**e**,**f**) signature proposed by Yang et al. [16] applied to CRC and NSCLC, respectively.

**Table 1 biomedicines-12-00703-t001:** Characteristics of patients with CRC, NSCLC, and controls.

Variable	CRC Group (N = 38)	NSCLC Group (N = 19)	Control Group (N = 20)	*p* Value (CRC vs. Control)	*p* Value (NSCLC vs. Control)
Age (Mean ± SD, years)	71.24 ± 12	72.79 ± 7.91	54.80 ± 14.97	<0.001 ^1,^*	<0.001 ^2,^*
Gender, N (%)				0.016 ^3,^*	0.265 ^3^
Male	24 (63.2)	9 (47.4)	6 (30)		
Female	14 (36.8)	10 (52.6)	14 (70)		
BMI group, N (%)				0.004 ^3,^*	0.008 ^3,^*
Normal weight (BMI < 25 kg/m^2^)	8 (21.1)	6 (31.6)	3 (15.0)		
Overweight (BMI ≥ 25 kg/m^2^ and <30 kg/m^2^)	20 (52.6)	9 (47.4)	3 (15.0)		
Obesity (BMI ≥ 30 kg/m^2^)	10 (26.3)	4 (21.1)	14 (70.0)		
Tumor location, N (%)					
Right colon	20 (52.6)				
Left colon	13 (34.2)				
Rectum	5 (13.2)				
TNM stage, N (%)					
I–II	17 (44.7)	13 (68.4)			
III–IV	21 (55.3)	6 (31.6)			

CRC: colorectal cancer; NSCLC: non-small cell lung cancer; SD: standard deviation; BMI: body mass index; ^1^ Mann–Whitney U test; ^2^ Student’s *t*-test; ^3^ Chi-squared test; * Statistically significant for the comparison at *p* value < 0.05.

**Table 2 biomedicines-12-00703-t002:** Differentially increased bacterial genera in feces from colorectal cancer (CRC) patients, non-small cell lung cancer (NSCLC) patients, and controls. (a) CRC vs. controls; (b) NSCLC vs. controls; (c) CRC vs. NSCLC.

**(a)**		**(b)**	
**Bacterial Genus**	***p* Values (LEfSe)**	**Bacterial Genus**	***p* Values (LEfSe)**
**Control group**			
*g__Sutterella*	0.019		
*g__Blautia*	0.005	**Control group**	
*g__Lachnoclostridium*	0.044	*g__Lachnoclostridium*	0.031
*g__Dielma*	0.018	*g__Megasphaera*	0.040
*g__Olsenella*	0.024	*g__Olsenella*	0.020
**CRC group**		**NSCLC group**	
*g__Campylobacter*	0.004	*g_Hungatella*	0.009
*g__Intestinimonas*	0.049	*g_Turicibacter*	0.013
*g__Parvimonas*	0.004	*g_Coprobacillus*	0.007
*g__Turicibacter*	0.006	*g_Clostridium sensu stricto 3*	0.015
*g__Hydrogenoanaerobacterium*	0.008	*g_Intestinimonas*	0.038
*g__Gemella*	0.016	*g_Eisenbergiella*	0.028
*g__Eisenbergiella*	0.008	*g_Salmonella*	0.015
*g__Clostridium sensu stricto 1*	0.021	*g_DTU089*	0.034
*g__Peptostreptococcus*	0.006	*g_Frisingicoccus*	0.040
*g__S5-A14a*	0.023	*g_Granulicatella*	0.035
*g__Ruminococcaceae*	0.022	*g_Incertae Sedis*	0.010
*g__Lactobacillus*	0.024	*g__Lactobacillus*	0.017
*g__Salmonella*	0.011	*g__Agathobacter*	0.038
*g__Eikenella*	0.034		
*g__CAG-352*	0.035		
*g__Fusobacterium*	0.003		
*g__Escherichia-Shigella*	0.018		
	**(c)**		
	**Bacterial Genus**	***p* Values (LEfSe)**	
	**NSCLC group**		
	*g__Bifidobacterium*	0.005	
	*g__Agathobacter*	0.004	
	*g__Roseburia*	0.050	
	*g__Blautia*	0.009	
	*g__CAG-873*	0.029	
	*g__Incertae Sedis*	0.039	
	*g__Oscillibacter*	0.005	
	*g__Ammoniphilus*	0.023	
	*g__Pseudobutyrivibrio*	0.005	
	**CRC group**		
	*g__Campylobacter*	0.030	
	*g__Ruminococcaceae*	0.042	
	*g__Hydrogenoanaerobacterium*	0.016	
	*g__Sellimonas*	0.021	
	*g__Solobacterium*	0.002	
	*g__Sarcina*	0.034	
	*g__Parvimonas*	0.005	
	*g__Peptostreptococcus*	0.003	
	*g__Eikenella*	0.033	
	*g__CAG-352*	0.037	
	*g__Fusobacterium*	0.004	

**Table 3 biomedicines-12-00703-t003:** Logistic regression analysis of bacterial genera remarked as increased in colorectal cancer (CRC) by LEfSe analysis, adjusting for sex, age, and body mass index (BMI).

CRC Bacterial Genera (Abundance Threshold)	Group	Unadjusted OR (95% CI)	Unadjusted *p* Value	Adjusted OR (95% CI)	Adjusted *p* Value
*Campylobacter* (0.003313)	Control	reference	-	reference	-
	Cancer	6.13 (1.72–21.9)	0.005 *	4.44 (0.88–22.4)	0.071
*Intestinimonas* (0.01827)	Control	reference	-	reference	-
	Cancer	2.35 (0.78–7.11)	0.130	1.34 (0.28–6.32)	0.705
*Parvimonas* (0.001859)	Control	reference	-	reference	-
	Cancer	10.9 (2.69–44.1)	0.001 *	53.3 (3.26–870.5)	0.005 *
*Turicibacter* (0.01017)	Control	reference	-	reference	-
	Cancer	6.13 (1.72–21.9)	0.005 *	3.98 (0.83–19.0)	0.084
*Hydrogenoanaerobacterium* (0.00367)	Control	reference	-	reference	-
	Cancer	4.56 (1.44–14.5)	0.010 *	1.42 (0.295–6.82)	0.662
*Gemella* (0.006006)	Control	reference	-	reference	-
	Cancer	3.68 (1.18–11.5)	0.025 *	6.01 (1.20–30.0)	0.029 *
*Eisenbergiella* (0.008012)	Control	reference	-	reference	-
	Cancer	4.56 (1.44–14.5)	0.010 *	5.35 (1.08–26.5)	0.040 *
*Clostridium sensu stricto 1* (0.08304)	Control	reference	-	reference	-
	Cancer	7.79 (1.95–31.2)	0.004 *	4.98 (0.93–26.6)	0.061
*Peptostreptococcus* (0.0003669)	Control	reference	-	reference	-
	Cancer	4.13 (1.24–13.7)	0.021 *	9.42 (1.38–64.2)	0.022 *
*S5-A14a* (0.0003232)	Control	reference	-	reference	-
	Cancer	6.55 (1.32–32.3)	0.021 *	2.54 (0.41–15.8)	0.318
*Ruminococcaceae* (0.004992)	Control	reference	-	reference	-
	Cancer	8.5 (2.19–33.0)	0.002 *	3.67 (0.62–21.7)	0.151
*Lactobacillus* (0.04309)	Control	reference	-	reference	-
	Cancer	5.67 (1.42–22.6)	0.014 *	6.72 (1.05–43.0)	0.044 *
*Salmonella* (0.00081)	Control	reference	-	reference	-
	Cancer	8.22 (2.40–28.2)	0.001 *	5.44 (1.02–28.8)	0.047 *
*Eikenella* (0.0001615)	Control	reference	-	reference	-
	Cancer	7.74 (0.92–65.1)	0.060	8.20 (0.73–92.1)	0.088
*CAG-352* (0.008875)	Control	reference	-	reference	-
	Cancer	3.42 (1.10–10.7)	0.034 *	2.14 (0.47–9.61)	0.323
*Fusobacterium* (0.01714)	Control	reference	-	reference	-
	Cancer	11.2 (3.02–41.6)	0.000 *	78.9 (4.48–1389.0)	0.003 *
*Escherichia_Shigella* (0.1569)	Control	reference	-	reference	-
	Cancer	5.2 (1.61–16.7)	0.006 *	3.81 (0.80–18.2)	0.094

CI: confidence interval; OR: odds ratio; * Statistically significant for the comparison at *p* value < 0.05.

**Table 4 biomedicines-12-00703-t004:** Logistic regression analysis of bacterial genera remarked as increased in non-small cell lung cancer (NSCLC) by LEfSe analysis, adjusting for sex, age, and body mass index (BMI).

NSCLC Bacterial Genera (Abundance Threshold)	Group	Unadjusted OR (95% CI)	Unadjusted *p* Value	Adjusted OR (95% CI)	Adjusted *p* Value
*Hungatella* (0.002103)	Control	reference	-	reference	-
	Cancer	6.53 (1.61–26.5)	0.009 *	7.17 (0.92–56.0)	0.060
*Turicibacter* (0.006865)	Control	reference	-	reference	-
	Cancer	5.20 (1.32–20.5)	0.019 *	4.61 (0.73–29.1)	0.104
*Coprobacillus* (0.0006056)	Control	reference	-	reference	-
	Cancer	6.86 (1.63–28.9)	0.009 *	7.00 (0.59–82.9)	0.123
*Clostridium sensu stricto 3* (0.0004242)	Control	reference	-	reference	-
	Cancer	5.50 (1.32–22.9)	0.019 *	6.11 (0.84–44.5)	0.074
*Intestinimonas* (0.0208)	Control	reference	-	reference	-
	Cancer	5.63 (1.36–23.3)	0.017 *	3.50 (0.45–27.3)	0.232
*Eisenbergiella* (0.007163)	Control	reference	-	reference	-
	Cancer	4.02 (1.06–15.3)	0.041 *	5.49 (0.69–43.9)	0.108
*Salmonella* (0.0008436)	Control	reference	-	reference	-
	Cancer	15.8 (2.80–89.0)	0.002 *	6.06 (0.73–50.4)	0.096
*DTU089* (0.007431)	Control	reference	-	reference	-
	Cancer	5.50 (1.32–22.9)	0.019 *	20.1 (1.35–300.1)	0.029 *
*Frisingicoccus* (0.0004159)	Control	reference	-	reference	-
	Cancer	5.06 (1.30–19.7)	0.020 *	2.88 (0.49–17.1)	0.244
*Granullicatella* (0.008393)	Control	reference	-	reference	-
	Cancer	4.02 (1.06–15.3)	0.041 *	4.81 (0.63–36.6)	0.129
*Incertae Sedis* (0.1568)	Control	reference	-	reference	-
	Cancer	8.00 (1.74–36.7)	0.007 *	160.1 (2.44–10,506.36)	0.017 *
*Lactobacillus* (0.03697)	Control	reference	-	reference	-
	Cancer	4.13 (1.06–16.1)	0.041 *	2.07 (0.36–11.8)	0.414
*Agathobacter* (0.5153)	Control	reference	-	reference	-
	Cancer	5.625 (1.36–23.3)	0.017 *	6.94 (0.97–49.6)	0.053

CI: confidence interval; OR: odds ratio; * Statistically significant for the comparison at *p* value < 0.05.

**Table 5 biomedicines-12-00703-t005:** Comparison of the diagnostic performance between the new proposed signatures for colorectal cancer (CRC) and non-small cell lung cancer (NSCLC) and two previously published signatures from large databases by test of equality of ROC areas for independent samples.

Panel	Bacterial Taxa Included	AUC (95% CI)	Se (95% CI)	Sp (95% CI)	*p* Value
CRC (proposed)	7 ^a^	0.840 (0.720–0.923)	78.9 (63.7–88.9)	80.0 (58.4–91.9)	-
Thomas et al. [15]	34 ^c^	0.557 (0.420–0.687)	60.5 (44.7–74.4)	60.0 (38.7–78.1)	0.004 *
Yang et al. [16]	43 ^d^	0.511 (0.376–0.644)	47.4 (32.5–62.7)	65.0 (43.3–81.9)	0.015 *
NSCLC (proposed)	2 ^b^	0.747 (0.583–0.872)	73.7 (51.2–88.2)	75.0 (53.1–88.8)	-
Thomas et al. [15]	34 ^c^	0.447 (0.288–0.615)	36.8 (19.1–59.0)	65.0 (43.3–81.3)	0.038 *
Yang et al. [16]	43 ^d^	0.413 (0.258–0.582)	47.4 (27.3–68.3)	45.0 (25.8–65.8)	0.002 *

AUC: area under the curve; CI: confidence interval; Se: sensitivity; Sp: specificity; * Statistically significant at *p* value < 0.05. ^a^ Proposed CRC panel: genera *Eisenbergiella*, *Fusobacterium*, *Gemella*, *Lactobacillus*, *Parvimonas*, *Peptostreptococcus* and *Salmonella*. ^b^ Proposed NSCLC panel: genera *DTU089* and *Ruminococcaceae Incertae Sedis*. ^c^ Signature by Thomas et al. [15]: *Acidaminococcus*, *Actinomyces*, *Anaerotruncus*, *Bilophila*, *Butyricimonas*, *Candidatus Schneewindia* (*CS*. *gallinarum*), *Campylobacter*, *Cloacibacillus*, *Dysosmobacter*, *Eggerthella*, *Eisenbergiella*, *Enterocloster*, *Enterococcus*, *Erysipelatoclostridium*, *Escherichia-Shigella*, *Faecalicatena*, *Faecalitalea*, *Flavonifractor*, *Hafnia-Obesumbacterium*, *Harryflintia*, *Holdemania*, *Hungatella*, *Hydrogeniiclostridium*, *Intestinimonas*, *Lachnoclostridium*, *Lancefieldela*, *Lawsonibacter*, *Merdimonas*, *Pseudoflavonifractor*, *Pararuminococcus*, *Ruthenibacterium*, *Scardovia*, *Streptococcus,* and *Veillonella*; ^d^ Signature by Yang et al. [16]: *Acidimicrobium*, *Acidiphilium*, *Actinomyces*, *Aeromonas*, *Alcaligenes*, *Anaplasma*, *Bacteroides*, *Bifidobacterium*, *Bradyrhizobium*, *Campylobacter*, *Capnocytophaga*, *Cellulomonas*, *Chamaesiphon*, *Eikenella*, *Flavobacterium*, *Fusobacterium*, *Gemella*, *Haemophilus*, *Halothiobacillus*, *Helicobacter*, *Hyphomicrobium*, *Leptotrichia*, *Methylophilus*, *Mycoplasma*, *Neisseria*, *Nostoc*, *Parvimonas*, *Pediococcus*, *Peptostreptococcus*, *Porphyromonas*, *Prevotella*, *Prosthecobacter*, *Pseudomonas*, *Roseiflexaceae—uncultured*, *Roseiflexus*, *Rothia*, *Rubrivivax*, *Selenomonas*, *Staphylococcus*, *Streptococcus*, *Treponema*, *Variovorax,* and *Veillonella*.

## Data Availability

We have deposited the raw sequence data in a public repository: Submission ID: SUB14179719. https://submit.ncbi.nlm.nih.gov/subs/sra/SUB14179719, accessed on 18 March 2024.

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
