# Peer review of "Gut Microbiota Signatures with Potential Clinical Usefulness in Colorectal and Non-Small Cell Lung Cancers"

_biomedicines, 2024, doi:10.3390/biomedicines12030703_

Round 1

Reviewer 1 Report

Comments and Suggestions for Authors

The authors highlight the use of bacterial metagenomic analysis as a biomarker for cancer detection, focusing on colorectal cancer (CRC) and non-small cell lung cancer (NSCLC). Analyzing 77 fecal samples from patients and controls, bacterial genomic sequencing revealed distinct microbiota profiles in CRC and NSCLC. After adjusting for confounders, specific bacteria were significantly associated with each cancer type. They claimed that their gut bacteria panels for CRC and NSCLC had better diagnostic capacity compared to existing panels, and show its potential application in cancer diagnosis. The study is very interesting, however, the experiment design is defective.

Major points:

1. Each graph in one figure should be mentioned and described properly.

2. P values in subject characteristics depicted in Table 1 should be included.

3. The author mentioned that “A total of 77 fecal samples from 38 CRC patients, 19 NSCLC patients and 20 Controls were collected and submitted to metagenomic analysis”, please include the calculation for sample size in the method.

4. Although the author mentioned that “The difference in age between the control group and the groups of patients affected by cancer could be, at least in part, a consequence of the exclusion criteria applied to subjects from the control group. It is not easy for an elderly patient not to meet any of the exclusion criteria that we have established (young patients meet the necessary characteristics more easily). Also in our case, mainly CRC is a pathology typically found in older patients, which probably explains the age differences” in the discussion, the huge differences in age between patients and control samples make the conclusion not credible.

Author Response

Responses to the Review Report (Reviewer 1):

1. Each graph in one figure has been mentioned and described properly, as reviewer suggested (in the revised version of the manuscript: lines 172-176, 178, 181, 183, 202-204, 216-220, 221, 235, 247, 285-286, 305-306 and 316-320).

2. P values in subject characteristics depicted in Table 1 have been now included (page 3).

3. In these studies, it may be difficult to calculate the sample size for the following reasons:

  • First of all, there is not a normal gut microbiota defined for the general population, since there are no universal relative abundances for the components of an unhealthy or healthy gut microbiota in control population. Therefore, sample size cannot be posited from a mean difference (e.g. if we could state that in the control population, the relative abundance of a particular genus is 0.25 and we aimed to detect a 20% increase -0.3- as significant, we could calculate a sample size, but these data are fictitious and knowledge about standard deviations is also lacking).
  • Secondly, data on diversity indexes to calculate a mean difference based on these parameters are lacking. Furthermore, in our study we do not find significant differences in diversity indexes.
  • On the other hand, the sample size could also be designed considering prevalence of dysbiosis in control population and the odds ratio of association between carcinogen factors and cancer. OR of association for environmental factors (such as tobacco, diet, toxics, viral agents…) has been previously defined and ranges from 1.2 to more. However, as far as we are concerned, dysbiosis has not been numerically defined, not even formally, (e.g. what is an imbalance of gut bacteria? An increase of 1, 2, 3… species? Which species?). Therefore, prevalence of dysbiosis cannot be appropriately calculated as prevalence of exposure factor for the risk of developing cancer.

The following references show how variable is the prevalence of “harmful” genera associated with different conditions, including cancer:

  • 9-fold higher Bilophila wadsworthia dsrA gene copy number in CRC patients than in controls. doi:10.1016/S0016-5085(15)30343-7.
  • Prevalence of B2 phylogroup coli harboring the colibatin-producing genes in biopsies of patients with CRC (55.3%) than in those of patients with diverticulosis (19.3%). PMID: 23457644.
  • Prevalence of Campylobacter by PCR in stool control samples for Bangladesh, Peru and Tanzania (63.9%, 42.7%, 72.5%).
  • 52% prevalence of pylori in asymptomatic controls. PMID: 2019355.
  • 16% of prevalence of positive PCR Fusobacterium in control tissue samples. PMID: 37312686.

With these data and using an OR of association between 1.2 to 4, sample size can range from 5748 to 80 (352 to 398 with OR = 2).

It is true that as of now, since we have our data, we could calculate sample size (we have relative abundance of hazardous genera and, as stated before, OR of other recognized carcinogens factors, e.g. >1.2).

Despite these limitations, we have added a comment in the Results section regarding the calculation of sample size and provide a putative sample size for future studies (lines 347-355, in the revised version of the manuscript).

4. In order to avoid bias in the results due to differences in age or other individual parameters, logistic regression was performed to adjust for potential confounding variables such as gender, age, and BMI differences (lines 159-161, in the Material and Methods section).

Reviewer 2 Report

Comments and Suggestions for Authors

In the manuscript submitted to me for review entitled "Gut microbiota signatures with potential clinical usefulness in colorectal and non-small cell lung cancers" the authors Sofía Tesolato, Juan Vicente-Valor, Mateo Paz-Cabezas, Dulcenombre Gómez-Garre, Silvia Sánchez-González, Adriana Ortega-Hernández, Sofía Cristina De la Serna, Inmaculada Domínguez-Serrano, Jana Dziakova, Daniel Rivera-Alonso, José Ramón Jarabo-Sarceda, Florentino Hernando Trancho, Ana maria gomez Martinez, Antonio José Torres-García, Pilar Iniesta propose a panel of gut bacteria that could be applied in the diagnosis of colorectal cancer (CRC) and non-small cell lung cancer (NSCLC).

            The study included a total of 77 fecal samples from patients diagnosed with CRC and NSCLC, as well as control volunteers. When conducting the experiments, all ethical norms were observed and the necessary consent was obtained from the patients.

            The methods used are well described. The obtained results are presented with the help of 6 figures and 5 tables and fully correspond to the conclusions made by the authors.

To support their research, the authors used 45 references that present information from studies spanning mostly the last decade. About 3/4 of the total references are from the last 5 years, indicating that the topic under review is relatively new and current and would be of interest to Biomedicines readers. I did not notice any redundant self-citations, all the references used are appropriate and necessary for the preparation of the manuscript.

My remarks and recommendations to the authors are:

1. On line 35 there is an inadvertent misspelling of the name Ruminoccocaeae (should be Ruminoccocaсеae). In the other places in the text, as far as I noticed, the same name is correctly written.

2. Under table 1, there are symbols representing the statistical methods used for the data presented in the table:

 "1Mann-Whitney U test; 2Student's T test; 3Chi-squared test; *Statistically significant for the comparison at p value < 0.05".

However, I do not see where in the table these designations are indicated. Let it be added in the appropriate places.

3. In the Back Matter of the manuscript before the References section, authors should add a Conflicts of interest section in which the authors declare that they have no conflict of interest (see instructions for authors).

4. In the References section, in a large part of the presented references, not all authors are presented (for example, in numbers 7, 8, 10, 13, 14, 15, 20, 21, 22, 23, 26, 27, 29, 31, 32, 34, 35, 37, 40, 41). Let's add all the authors - in my opinion this would benefit the readers of the manuscript.

Author Response

Responses to the Review Report (Reviewer 2):

  1. Ruminoccocaeae has been corrected to Ruminococcaсеae in line 35 and in other lines where it was found to be misspelled (lines 315, 397 and 407).
  2. Table 1 (page 3) has been completed including P values of subject characteristics.
  3. Conflicts of interest section is now included in the manuscript (line 480).
  4. All the authors of each of the references included in the manuscript have now been included (References section, pages 19-21).

Round 2

Reviewer 1 Report

Comments and Suggestions for Authors

Looks good!